# A Critical Analysis of the Robustness of Radiomics to Variations in Segmentation Methods in ^18^F-PSMA-1007 PET Images of Patients Affected by Prostate Cancer

**DOI:** 10.3390/diagnostics13243640

**Published:** 2023-12-11

**Authors:** Giovanni Pasini, Giorgio Russo, Cristina Mantarro, Fabiano Bini, Selene Richiusa, Lucrezia Morgante, Albert Comelli, Giorgio Ivan Russo, Maria Gabriella Sabini, Sebastiano Cosentino, Franco Marinozzi, Massimo Ippolito, Alessandro Stefano

**Affiliations:** 1Department of Mechanical and Aerospace Engineering, Sapienza University of Rome, Eudossiana 18, 00184 Rome, Italy; giovanni.pasini@uniroma1.it (G.P.); morgantelucrezia@gmail.com (L.M.); franco.marinozzi@uniroma1.it (F.M.); 2Institute of Molecular Bioimaging and Physiology, National Research Council (IBFM-CNR), Contrada, Pietrapollastra-Pisciotto, 90015 Cefalù, Italy; giorgio.russo@ibfm.cnr.it (G.R.); richiusaselene3@gmail.com (S.R.); acomelli@fondazionerimed.com (A.C.); alessandro.stefano@ibfm.cnr.it (A.S.); 3National Laboratory of South, National Institute for Nuclear Physics (LNS-INFN), 95125 Catania, Italy; 4Nuclear Medicine Department, Cannizzaro Hospital, 95125 Catania, Italy; cmantarro@gmail.com (C.M.); ianocose@hotmail.com (S.C.); ippolitomas@yahoo.it (M.I.); 5Ri.MED Foundation, Via Bandiera 11, 90133 Palermo, Italy; 6Department of Surgery, Urology Section, University of Catania, 95125 Catania, Italy; giorgioivan.russo@unict.it; 7Medical Physics Unit, Cannizzaro Hospital, 95125 Catania, Italy; mgabsabini@gmail.com

**Keywords:** radiomics, prostate, machine learning, ^18^F-PSMA-1007 PET, segmentation, robustness, reproducibility

## Abstract

Background: Radiomics shows promising results in supporting the clinical decision process, and much effort has been put into its standardization, thus leading to the Imaging Biomarker Standardization Initiative (IBSI), that established how radiomics features should be computed. However, radiomics still lacks standardization and many factors, such as segmentation methods, limit study reproducibility and robustness. Aim: We investigated the impact that three different segmentation methods (manual, thresholding and region growing) have on radiomics features extracted from ^18^F-PSMA-1007 Positron Emission Tomography (PET) images of 78 patients (43 Low Risk, 35 High Risk). Segmentation was repeated for each patient, thus leading to three datasets of segmentations. Then, feature extraction was performed for each dataset, and 1781 features (107 original, 930 Laplacian of Gaussian (LoG) features, 744 wavelet features) were extracted. Feature robustness and reproducibility were assessed through the intra class correlation coefficient (ICC) to measure agreement between the three segmentation methods. To assess the impact that the three methods had on machine learning models, feature selection was performed through a hybrid descriptive-inferential method, and selected features were given as input to three classifiers, K-Nearest Neighbors (KNN), Support Vector Machines (SVM), Linear Discriminant Analysis (LDA), Random Forest (RF), AdaBoost and Neural Networks (NN), whose performance in discriminating between low-risk and high-risk patients have been validated through 30 times repeated five-fold cross validation. Conclusions: Our study showed that segmentation methods influence radiomics features and that Shape features were the least reproducible (average ICC: 0.27), while GLCM features the most reproducible. Moreover, feature reproducibility changed depending on segmentation type, resulting in 51.18% of LoG features exhibiting excellent reproducibility (range average ICC: 0.68–0.87) and 47.85% of wavelet features exhibiting poor reproducibility that varied between wavelet sub-bands (range average ICC: 0.34–0.80) and resulted in the LLL band showing the highest average ICC (0.80). Finally, model performance showed that region growing led to the highest accuracy (74.49%), improved sensitivity (84.38%) and AUC (79.20%) in contrast with manual segmentation.

## 1. Introduction

Prostate cancer (PCa) is estimated to be the most prevalent type of cancer in men in 2023 and the second leading cause of cancer death, only below lung and bronchus cancer [1]. Its early diagnosis is commonly carried out through prostate specific antigen (PSA) screening, followed up by transrectal ultrasound-guided biopsy (TRUS-biopsy) in men who present elevated values of PSA. However, PSA testing includes poor specificity, thus leading to patients over treatment and men undergoing unnecessary biopsies [2,3]. Moreover, prostate biopsy is associated with some risks, such as hematuria, infection, pain, inflammation and sepsis [4]. Therefore, imaging modalities such as Magnetic Resonance Imaging (MRI) have been introduced to support the diagnostic process, providing accurate prostate cancer localization, guiding targeted biopsies and for local cancer staging [5]. However, MRI could not be sufficiently accurate in detecting certain portions of prostate cancer lesions, that only by adding molecular information provided by Positron Emission Tomography (PET) imaging could be identified [6]. Moreover, PET imaging showed excellent sensitivity and specificity for recurrent prostate cancer and promising results in the detection of bone metastasis, especially when the Prostate-specific membrane antigen (PSMA) is targeted by radioligands such as ^68^Ga (Gallium 68) [7,8]. However, its shortcomings related to short-half life, non-ideal energies and difficult production motivated the consideration of ^18^F-labelled analogs, and the ^18^F-PSMA-1007 has been individuated as the candidate compound to overcome such issues [9].

Furthermore, radiomics has recently emerged as a promising technique harnessing advanced computational methodologies to derive quantitative data from medical images, such as PET [10], MRI [11], Computed Tomography (CT) [12], or molecular hybrid imaging [13]. These quantitative metrics are subsequently employed for constructing predictive models that can offer assistance in diagnosing, planning treatment and forecasting outcomes across a spectrum of diseases, spanning from oncology [14] to neurodegenerative disorders [15,16].

Regarding prostate cancer, PET-based radiomics has been successfully used to predict intraprostatic lesions [17,18], the low-vs-high lesion risk [19], the Gleason Score (GS) [20] and bone metastases [21,22]. However, most of the studies employed ^68^Ga-PSMA-11 [17,18], while few radiomics applications investigated the potential of ^18^F-PSMA-1007 [20,23]. Moreover, although its several benefits and the publication of the Imaging Biomarker Standardization Initiative (IBSI), that established how radiomics features should be calculated [24], radiomics still lacks standardization and the variability in segmentation methods, image pre-processing parameters and machine learning pipeline could limit study reproducibility [25]. In the literature, there are several studies that investigated the impact of segmentation methods on radiomics features [26,27,28], but none focused on ^18^F-PSMA-1007 PET-based images of patients affected by prostate cancer. Therefore, the aim of this study is to focus on the impact of segmentation methods on ^18^F-PSMA-1007 PET-based radiomics models for the prediction of the high pathological grade in prostate cancer and on the robustness and reproducibility of radiomics features to variations in segmentation methods. Furthermore, this study highlights the potential of ^18^F-PSMA-1007 PET in differentiating between high-risk and low-risk prostate cancer.

## 2. Materials and Methods

### 2.1. Study Design

Eighty-one patients underwent ^18^F-PSMA-1007 PET/CT imaging using two different scanners (General Electric Milwaukee, WI, USA, Discovery 690FX&MOT and Siemens Knoxville, TN, USA, Biograph Horizon 4R). For the first scanner (30 patients), 16 row helical CT scan was used with the following conditions: tube voltage (140 kVp), tube current (800 mA_max_). PET collection time of every bed was 90 s, the whole-body scanning needed 7–8 beds. PET matrix was 256 × 256, and CT matrix was 512 × 512. PET voxel size was 2.73 × 2.73 × 3.27 mm^3^ and CT voxel size was 1.37 × 1.37 × 3.75 mm^3^. For the second scanner (51 patients), 16 row helical CT scan was used with the following conditions: tube voltage (130 kVp), tube current (345 mA_max_). PET collection time of every bed was 90 s, the whole-body scanning needed 6–7 beds. PET matrix was 512 × 512, and CT matrix was 512 × 512. The PET voxel size was 1.45 × 1.45 × 3 mm^3^ and the CT voxel size was 0.98 × 0.98 × 3 mm^3^. The tracer produced was intravenously injected into patients at a standardized dose of 4 MBq/kg. After the tracer administration, the patients rested in a quiet room for about 120 min before scanning.

Inclusion patients’ criteria were: (1) diagnosis of PCa through biopsy; (2) elevated serum PSA value; (3) no therapy before PET scan, neither surgery, chemotherapy, radiotherapy, endocrine therapy, or anything else. Subjects who met the above three criteria simultaneously were included in our study. The patients without exhibiting relevant radiotracer uptake above background in the prostate on ^18^F-PSMA-1007 imaging were excluded. Consequently, at least in this phase, only patients with positive PET scans were considered. The study complies with the Declaration of Helsinki, and local ethics committee approval was obtained (REDIRECT Study, n.101/2022). Finally, the number of patients was further decreased to 78 (29 GE Milwaukee, WI, USA, 49 Siemens Knoxville, TN, USA) due to segmentation reasons explained in Section 2.3. The complete tables are reported in Section 3.1.

### 2.2. Gleason Score

PCa often demonstrates varying levels of aggressiveness across different regions of the tumor due to its inherent heterogeneity [29]. To quantify this variation, a grading system is used to assign grades to the two primary areas of cancerous tissue under examination, and the sum of these grades determines the overall Gleason score (GS). In other words, the GS is a scale used to assess the severity of prostate cancer based on its biopsy. The GS ranges from 6 to 10 and is assigned based on the appearance of cancer cells in tissue samples. The higher the GS, the greater the severity of prostate cancer. Specifically, the pathologist identifies the two most prevalent patterns of cancerous growth within the tissue samples. Each pattern is assigned a grade on the GS, ranging from 1 to 5. The two grades assigned to the patterns are added together to obtain the final GS. The first number represents the predominant grade within the tumor. For instance, a GS of 3 + 4 = 7 signifies that the tumor is predominantly grade 3, with a smaller portion being grade 4. These two grades are summed to derive the final GS, which, in this instance, is 7. In general, the degree of aggressiveness in PCa can be categorized as follows [30]:GS ≤ 6: Signifying tumors with slow growth tendencies that typically do not metastasize to distant organs beyond the prostate (non-metastatic).GS = 7: Indicating tumors of intermediate aggressiveness.GS between 8 and 10: Corresponding to highly aggressive tumors with a propensity for metastasis.

In our study, tumors with GS 3 + 3 or 3 + 4 were classified as low grade, while tumors with GS 4 + 3, 4 + 4, or 4 + 5 were designated as high grade.

### 2.3. Segmentation and Segmentation Agreement

For each patient, image segmentation was performed through both manual and semi-automated methods to extract PCa volumes from PET studies once uploaded to matRadiomics 1.5 [31], a comprehensive radiomics framework that enables the import of biomedical images, the segmentation of the target, the extraction and selection of the radiomics features and the implementation of the predictive model via machine learning algorithms within the same software. Regarding the segmentation task, for the manual approach, an experienced nuclear medicine physicist (C.M. author) manually performed the slice-by-slice delineation. For the semi-automatic approach, we used two semi-automatic segmentation algorithms implemented in matRadiomics 1.5 [31], namely region growing [32,33] and thresholding [34,35]. Specifically, both semi-automatic methods were initialized drawing a Region of Interest (ROI) that surrounds the target and locates the portion of the image that was subsequentially elaborated by the region growing algorithm, based on the “*activecontour*” MATLAB function [32,33], and the thresholding algorithm [34,35], that uses a percentage of the maximum level of grey in the ROI as a threshold. Since three patients could not be segmented through region growing and thresholding, they were discarded from the total, that decreased from 81 to 78. Figure 1 shows the workflow that led from image visualization to image segmentation. Finally, we compared the obtained binary segmentations using the Jaccard Index [36] given in the equation below with *X* and *Y* being sets of segmentations.
(1)$$J\left(X,Y\right)=\frac{\left|X\cap Y\right|}{\left|X\cup Y\right|}=\frac{\left|X\cap Y\right|}{\left|X\right|+\left|Y\right|-\left|X\cap Y\right|}$$

### 2.4. Feature Extraction

Feature extraction was repeated, for each patient, using the three sets of segmentations (manual, based on region growing and based on thresholding), ending with three datasets of extracted features. A total of 1781 radiomics features were extracted per dataset using the Pyradiomics [37] extractor integrated in the matRadiomics software 1.5. Radiomics features were extracted from the original images (107), from the wavelet decomposed images (744) and from the Laplacian of Gaussian (LoG) filtered images (930), and they could be grouped in three categories: (i) Shape features, based on target morphology, (ii) First Order Statistics Features, based on the distribution of level of grays within the target, and (iii) Texture features based on the pattern of level of grays within the target. Furthermore, Texture features could be grouped in the gray level co-occurrence matrix (GLCM), gray level run length matrix (GLRLM), gray level size zone matrix (GLSZM), neighboring gray tone difference matrix (NGTDM) and the gray level dependence matrix (GLDM) [38,39,40,41,42].

The Pyradiomics configuration used in our analysis is reported in Table 1. The other parameters were left to Pyradiomics’ default. Figure 2 shows the workflow from feature segmentation to feature extraction.

### 2.5. Feature Robustness

After feature extraction, the intraclass correlation coefficient (ICC) was calculated for every feature to quantify inter-observer feature reproducibility and consequently feature robustness when using three segmentation methods. The ICC score ranges from 0 to 1, representing no reproducibility to perfect reproducibility, respectively. Following Koo and Li guidelines [43], ICC values have been grouped in ranges, ICC < 0.5, 0.5 < ICC < 0.75, 0.75 < ICC < 0.9, ICC > 0.9, indicating poor, moderate, good and excellent reproducibility, respectively. It was computed using the formula proposed by McGraw and Wong in case 3A (A,1) [44] to measure absolute agreement as
(2)$$ICC=\frac{MS_{R}-MS_{E}}{MS_{R}+\left(k-1\right)MS_{E}+\frac{k}{n}\left(MS_{C}-MS_{E}\right)}$$
where *MS_R_* = mean square for rows, *MS_E_* = mean square error, *MS_C_* = mean square for columns, *k* = number of observers involved and *n* = number of subjects, *j* indicates the *j*-th feature and *K_Class_* the total number of features for each class.

The ICCs were consequently averaged by grouping features by feature *Class*, thus obtaining the ICC_Shape_, ICC_Statistics_, ICC_GLCM_, ICC_GLDM_, ICC_GLRLM_, ICC_GLSZM_, ICC_NGTDM_. The general formula is
(3)$$ICC_{Class}=\frac{1}{K_{Class}}\sum_{j=1}^{K_{Class}}ICC_{j}$$

Finally, boxplots that summarize the distributions of ICC values were derived, as detailed in Section 3, specifically within Section 3.3.

### 2.6. Feature Selection and Machine Learning Methodology

For each dataset of extracted features, feature selection was performed through a hybrid descriptive-inferential approach to streamline the feature reduction and selection process. This method uses point biserial correlation to assign scores to the features, arrange them in descending order based on their scores and then iteratively construct a logistic regression model, as extensively documented in [45]. To sum it up, during each iteration, the model’s *p*-value is compared to the previous iteration’s *p*-value. If the *p*-value fails to decrease in the current iteration, the process concludes and the logistic regression model is established. We ended with three subsets of selected features (manual, region growing and thresholding) that were given as input to six machine learning models, Linear Discriminant Analysis (LDA), K-Nearest Neighbors (KNN), Radial Basis Function (RBF) Support Vector Machines (SVM), Random Forest (RF), AdaBoost and Neural Networks (NN). Models’ performances were assessed through 30 times repeated five-fold cross validation and accuracy, area under curve (AUC), sensitivity, specificity, precision and f score were calculated. Cross validation was implemented in such a way that each model was trained and validated with the same folds.

### 2.7. Statistical Analysis

Data were first analyzed through Lilliefors test [46,47], histogram visual inspection and q-q plots (quantile-quantile plots) to verify the assumption of normality and using Levene’s test [48] to assess homogeneity of variance. Therefore, Kruskal–Wallis test [49,50] was used to assess if feature classes (Shape, First Order Statistics, GLCM, GLRLM, GLSZM, GLDM, NGTDM) had a statistically significant impact on ICC values for each image type (original, LoG, wavelet), Friedman test [51,52] was used to assess if wavelet sub-bands (HHH, HLH, HHL, HLL, LLL, LHL, LLH, LHH) had a statistically significant impact on wavelet features. Moreover, to assess if models and segmentations had a statistically significant impact on model performance, two separate Friedman tests [26,51,52] were performed. Each test was followed by a post hoc test corrected with Dunn–Sidak [53] correction for multiple comparisons. Finally, 95% confidence intervals were calculated for each model performance through 1000 bootstrapping [54].

Statistical analysis was performed using MATLAB R2023b [55].

## 3. Results

### 3.1. Clinical Data

A total of 81 patients were included in this study and they were divided into two groups based on their GS score (46 Low Grade, 35 High Grade) as reported in Table 2.

After segmentation, because the semi-automated algorithms failed to correctly identify the tumor region in 3 patients, the total number of patients decreased to 78 (43 Low Grade, 35 High Grade), as shown in Table 3. Specifically, three low-grade GS patients were removed from the analysis.

### 3.2. Segmentation Agreement

The segmentations obtained using manual, thresholding (TS) and region growing (RG) methods were compared pairwise through the Jaccard Index. The lowest average Jaccard Index value (0.51) was obtained comparing manual and RG segmentations, while the highest average Jaccard Index value (0.58) was obtained comparing TS and RG methods, as shown in Figure 3.

### 3.3. Feature Robustness Results

After feature extraction, feature robustness was assessed and the results are summarized in the box plots shown in Figure 4, Figure 5 and Figure 6. Each class of features is represented by a boxplot, together with the average ICC value, and each figure is representative of the images used (original, LoG filtered and wavelet decomposed). For the original images, the GLCM feature class reached the highest average ICC value (ICC_GLCM-original_ = 0.89), while the Shape feature class reached the lowest (ICC_Shape-original_ = 0.27). The GLSZM feature class reached the second lowest average ICC value (ICC_GLSZM-original_ = 0.65). For the LoG filtered images, the GLCM feature class reached the highest average ICC value (ICC_GLCM-LoG_ = 0.87), while the GLSZM feature class reached the lowest (ICC_GLSZM-LoG_ = 0.72). For the wavelet decomposed images, the GLCM and the First Order Statistics feature classes reached the highest average ICC values (ICC_GLCM-wavelet_ = 0.56, ICC_First Order Statistics-wavelet_ = 0.56), while the GLSZM feature class reached the lowest (ICC_GLSZM-wavelet_ = 0.44). Complete results are reported in Table 4. Statistical significance between feature class groups was assessed using the Kruskal–Wallis analysis and was repeated for each image type (original, log and wavelet). Since the Kruskal–Wallis tests were significant (p_threshold_ = 0.05) for each image type group (p_original_ = 3.5 × 10^−7^, p_LoG_ = 4.3 × 10^−12^, p_wavelet_ = 0.0031), they were followed by post hoc tests corrected by the Dunn–Sidak correction for multiple comparisons. For the original image type, the ICCs of the Shape feature class differed significantly from those belonging to the First Order Statistics (p_Shape-Statistics_ = 2.1 × 10^−6^), GLCM (p_Shape-GLCM_ = 3.6 × 10^−7^) and GLRLM (p_Shape-GLRLM_ = 0.02) feature classes. For the LoG image type, the ICCs of both the First Order Statistics and GLCM feature classes differed significantly from those belonging to the GLSZM (p_Statistics-GLSZM_ = 0.00012, p_GLCM-GLSZM_ = 2.1 × 10^−5^), GLDM (p_Statistics-GLDM_ = 1.2 × 10^−5^, p_GLCM-GLDM_ = 1.8 × 10^−6^) and NGTDM (p_Statistics-NGTDM_ = 4.2 × 10^−5^, p_GLCM-NGTDM_ = 1.7 × 10^−5^) feature classes. In addition, the ICCs values belonging to the GLDM and NGTDM feature classes differed significantly from those belonging to the GLRLM class (p_GLDM-GLRLM_ = 0.014, p_NGTDM-GLRLM_ = 0.0057). For the wavelet image type, the ICC values of both the First Order Statistics and GLCM feature classes differed significantly from those belonging to the GLSZM (p_Statistics-GLSZM_ = 0.0047, p_GLCM-GLSZM_ = 0.0034) feature class.

As reported in Figure 7, the following results were obtained:Excellent reproducibility (ICC > 0.9) was reached by 51.18% LoG features, 48.60% original features and only 9.01% wavelet features.Good reproducibility (0.75 < ICC < 0.9) was reached by 14.02% original features, 22.47% LoG features and 17.07% wavelet features.Moderate reproducibility (0.5 < ICC < 0.75) was reached by 18.69% original features, 15.38% LoG features, 26.08% wavelet features.Poor reproducibility (ICC < 0.5) was reached by 18.69% original features, 10.97% LoG features, 47.85 % wavelet features.

**Figure 7 diagnostics-13-03640-f007:**
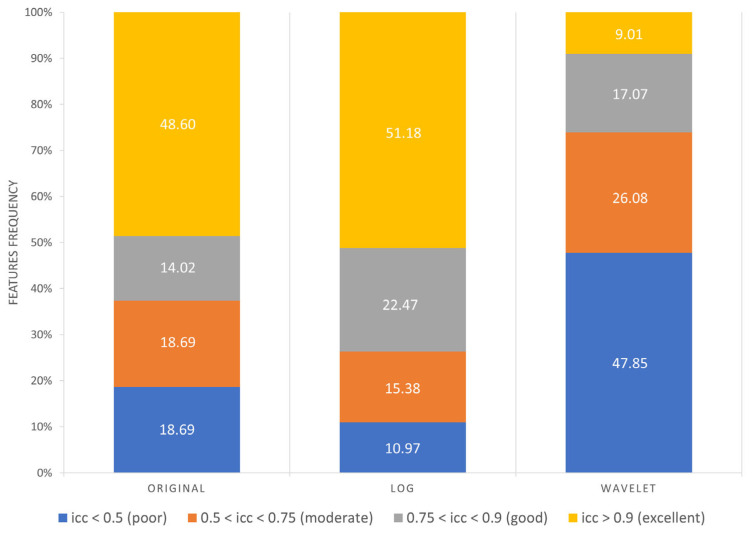
Frequency of poor, moderate, good, excellent reproducible features divided by image type.

Furthermore, as shown in Figure 8, 71.43% Shape features obtained poor reproducibility, and only 7.14% Shape features reached good reproducibility, while excellent reproducibility was reached by 38.30% First Order features, 42.98% GLCM features, 23.31% GLDM features, 32.57% GLRLM features, 28.95% GLSZM features and 20% NGTDM features. Finally, ICC values were evaluated per wavelet band (HHH, HHL, HLH, HLL, LLL, LLH, LHL, LHH), as shown in Figure 9.

The highest average ICC (0.8) was reached by the LLL sub-band, while the lowest average ICC (0.34) was reached by the HHL sub-band.

Since the Friedman test suggested statistical significance (p_threshold_ = 0.05) between sub-bands groups, a post hoc test with Dunn–Sidak correction for multiple comparisons was performed. It resulted that the ICCs of the LLL sub-band significantly differed from those belonging to the other sub-bands (p_LLL-LHL_ = 1.1 × 10^−8^, p_LLL-LLH_ = 5.0 × 10^−8^, p_LLL-LHH_ = 6.3 × 10^−5^, p_LLL-HHH_ << 0.05, p_LLL-HHL_ << 0.05, p_LLL-HLH_ << 0.05, p_LLL-HLL_ << 0.05). Moreover, the LHL, LHH and LLH sub-band ICC values differed significantly from those belonging to HHH (p_LHL-HHH_ = 5.6 × 10^−11^, p_LHH-HHH_ << 0.05, p_LLH-HHH_ = 9.8 × 10^−12^), HLH (p_LHL-HLH_ = 0.0010, p_LHH-HLH_ = 4.3 × 10^−7^, p_LLH-HLH_ = 0.00035), HHL (p_LHL-HHL_ << 0.05, p_LHH-HHL_ << 0.05, p_LLH-HHL_ << 0.05) and HLL (p_LHL-HLL_ = 4.19 × 10^−9^, p_LHH-HLL_ = 5.9 × 10^−14^, p_LLH-HLL_ = 8.5 × 10^−10^) sub-bands. In addition, the ICC values of the HLH sub-band differed significantly from those belonging to the HHL (p_HLH-HHL_ = 3.0 × 10^−5^) sub-band. The notation *p* << 0.5 is used instead of *p* = 0, obtained when the *p* was very low and software precision could not represent it.

### 3.4. Feature Selection and Machine Learning Results

After feature extraction, feature selection was performed on the three datasets of extracted features (manual, region growing, thresholding). We ended with three subsets of selected features, all containing only one feature, as shown in Table 5. For the dataset based on manual segmentation, only the feature named “*wavelet_LLL_firstorder_Minimum*” was selected, while the feature named *“wavelet_HHL_glszm_LowGrayLevelZoneEmphasis*” was selected for the dataset based on thresholding segmentation. For the dataset based on region growing segmentation, the feature named “*wavelet_HLH_glszm_LowGrayLevelZoneEmphasis*” was selected. All selected features belonged to the wavelet decomposed image type. Six machine learning models based on the three subsets of selected features were built using LDA, KNN, SVM, RF, AdaBoost and NN, whose performances (accuracy, AUC, sensitivity, specificity, precision, fscore) are reported in Table 6, Table 7, Table 8, Table 9, Table 10 and Table 11. All performances are averaged on 30 repetitions of five-fold cross validation. Results show that all six models based on manual segmentation reached the highest specificity (LDA: 86.05%, SVM: 81.40%, KNN: 79.61%, RF: 75.66%, AdaBoost: 70.23%, NN: 68.91%) and the highest precision (LDA: 75.10%, SVM: 71.43%, KNN: 70.19%, RF: 65.90%, AdaBoost: 59.87%, NN: 54.69%) when compared to the models based on region growing (RG) and thresholding (TS). Instead, sensitivity was always higher in the models based on semi-automatic segmentation methods, and the RG-SVM reached the highest value (sensitivity = 85.24%), while the RG-LDA had the second highest value (sensitivity = 84.38%). Moreover, all six models based on region growing segmentation reached the highest fscore (LDA: 74.80%, SVM: 73.09%, KNN: 71%, RF: 70.54%, AdaBoost: 61.29%, NN: 64.29%) when compared to the models based on manual and thresholding segmentations.

The RG-LDA also reached the highest accuracy (74.49%) and the highest AUC (79.20%).

Finally, even if the Friedman test suggested that models had a statistically significant impact on machine learning performance, post hoc tests corrected with Dunn–Sidak correction for multiple comparisons did not find a significant performance difference between models. On the contrary, the Friedman test suggested that segmentation had a statistically significant impact on model performance, and statistical significance was also found during post hoc tests corrected with the Dunn–Sidak correction for multiple comparisons. It resulted that accuracies differed significantly (p_RG-TS_ = 0.022) between region growing and thresholding methods, that AUCs differed significantly (p_manual-RG_ = 0.0067, p_TS-RG_ = 0.031) between manual and region growing methods, and between thresholding and region growing methods, that sensitivities differed significantly (p_manual-RG_ = 0.0012) between manual and region growing methods, that specificities differed significantly between manual and region growing (p_manual-RG_ = 0.032) methods and between manual and thresholding methods (p_manual-TS_ = 0.0077), that precisions differed significantly (p_manual-TS_ = 0.027) between manual and thresholding methods, and that the fscore differed significantly between manual and region growing methods (p_manual-RG_ = 0.0061). Complete results per iteration are provided for each model and segmentation methods in the Appendix A.

## 4. Discussion

It has been shown in several studies that radiomics has the potential to support the clinical decision-making process, but it still lacks standardization, and several factors such as image pre-processing, image kernel reconstruction settings and segmentation methods could influence radiomics feature values [56]. Concerning prostates, most of the studies employed MRI images rather than PET images, and only few of them investigated radiomics feature reproducibility and robustness. In MRI images, it has emerged that normalization techniques and manual segmentation repeated by different operators negatively impact radiomics feature reproducibility [57,58]. Indeed, manual segmentation is operator-dependent, as shown in several studies not only related to prostates, and semi-automatic methods should be preferred to improve radiomics study reproducibility [59,60,61]. Therefore, we analyzed the effect of three segmentation methods (manual, thresholding, region growing) before and after feature extraction. The Jaccard Index showed that there were dissimilarities between segmentations, reaching the maximum value of 0.58 when comparing thresholding and region growing methods. From our study, it emerged that Shape features obtained the lowest average ICC (0.27), and 71.43% of Shape features obtained poor reproducibility (ICC < 0.5). This behavior could be due to the difficulties in identifying target contours in PET images, and it is in line with [62], that showed that Shape features obtained the lowest ICC performance, even if higher than those obtained in our study, but radiomics features were extracted with a different software, different radiotracers were used and the clinical aim was different. Otherwise, Texture features performed better, with GLCM features exhibiting the highest average ICCs value in all the three image types (original, LoG and wavelet), with 61.4% GLCM features obtaining good (18.42%) and excellent (42.98%) reproducibility. Our study also shows that reproducibility differs depending on image type, and therefore on image filtering. Indeed, LoG features exhibited a higher rate of features with excellent reproducibility (51.18%) when compared with original features (48.60%) and wavelet features (9.01%). However, comparison with original features is not completely accurate since LoG features and wavelet features do not include the Shape feature class, only present in the original features. Moreover, the analysis showed that wavelet decomposition is the worst performing filter in terms of reproducibility, with 47.85% wavelet features exhibiting poor reproducibility. It has been shown in previous studies [63,64] that image denoising could lead to more robust features, and the difference in performance between LoG and wavelet could be due to the different wavelet sub-band combinations. Indeed, the combinations that started with a high pass filter (HHH, HLH, HHL, HLL) obtained the lowest average ICC (ICC_HHL_: 0.34), while those that started with a low pass filter obtained the highest values, with the LLL sub-band showing the highest average ICC (0.8). This could be since low pass filtering managed to suppress image noise.

In addition, we showed that different segmentation methods lead to different subsets of selected features. However, feature selection based on semi-automatic methods led to the same selected feature, namely “*glszm_LowGrayLevelZoneEmphasis*”, even if processed with a different wavelet combination. This result could be since semi-automatic methods are more reproducible and similar when compared with manual segmentations. To assess if thresholding and region growing are more reproducible methods than manual segmentation, inter-observer variability per single method should be evaluated. Furthermore, machine learning models were impacted by segmentation methods. All manual-based models showed higher specificity than sensitivity, with a maximum of 86.05% reached by the LDA classifier, while semi-automatic methods showed higher sensitivity than specificity, with a maximum of 85.24% reached by the RG-SVM classifier, while the RG-LDA classifier obtained the second highest sensitivity (84.38%). Moreover, statistical tests suggested that the model type did not have a significantly impact on model performance and differences between performance derived mainly from segmentation methods. From our results, it seems that manual segmentation is more specific rather than sensitive, and therefore there are radiomics features that are more sensitive and/or specific than others. Statistical tests also corroborated the findings that region growing-based models enhanced sensitivities, AUCs and accuracies when compared with manual-based models.

Finally, our study demonstrates that ^18^F-PSMA-1007 PET radiomics can differentiate between low-risk and high-risk prostate patients, reaching the highest accuracy (74.49%) with the RG-LDA model (accuracy: 74.49%, AUC: 79.20%, sensitivity: 84.38%). However, models were based on wavelet features that showed the worst average ICC performance (ICC_wavelet-Statistics_ = 0.56, ICC_wavelet-GLSZM_ = 0.44), affecting the ability of the model to generalize.

All our analyses have been based on matRadiomics 1.5 [31], which emerges as a tool in addressing the imperative of explainability in radiomics. Explainability is a critical aspect that enhances the interpretability of complex models and fosters trust in their outcomes [65]. matRadiomics 1.5, with its comprehensive suite of functionalities, provides a robust platform to segment biomedical images and extract and analyze a wide array of quantitative features, allowing users to complete the whole radiomics workflow within a single software, simplifying the process and making it replicable. It not only has the capacity to generate radiomics models but also to offer transparency and interpretability. As machine learning algorithms play an increasingly pivotal role in radiomics, understanding how these models arrive at specific predictions is essential for clinical acceptance. Clinicians and healthcare professionals require insights into the decision-making processes of these algorithms, especially when the stakes involve patient diagnosis and treatment planning. Transparent and interpretable radiomics models facilitate a deeper understanding of the features influencing predictions. Efforts in research and development are directed towards creating machine learning models in radiomics that not only deliver accurate results but also provide interpretable rationales for their predictions, ultimately contributing to improved patient care and diagnostic precision.

To date, this is the first study based on ^18^F-PSMA-1007 PET imaging of patients affected by prostate cancer, with the aim of evaluating the robustness and reproducibility of radiomics features to variations in segmentation methods and the impact that segmentation methods have on selected features and model performance, thereby contributing to the standardization of radiomics. However, this study has limitations, such as the small sample size and its single-center nature that could affect results generalization. In fact, features robustness and reproducibility depend on the dataset used, even if the clinical aim is the same, and/or from the center that acquired the images [66], and multicenter studies should be preferred. This issue can also be extended to the models’ generalization ability. However, the main goal of this study was not to build the best machine learning models but to study the effects of segmentation methods on model performance.

## 5. Conclusions

The present study, which utilizes ^18^F-PSMA-1007 PET imaging of prostate cancer patients, investigated the robustness and reproducibility of radiomics features to variations in segmentation methods. Additionally, it examined the influence that segmentation methods had on selected features and model performance. The results showed that the Shape feature class is the least robust, while the GLCM feature class is the most robust. In addition, segmentation methods impacted feature selection, resulting in a higher specific feature when manual segmentation was used and in higher sensitive features when semi-automatic methods were used. Finally, our study demonstrates that ^18^F-PSMA-1007 PET radiomics using the RG-LDA model can differentiate between low-risk and high-risk prostate patients, reaching the best performance (accuracy: 74.49%, AUC: 79.20%, sensitivity: 84.38%)

## Figures and Tables

**Figure 1 diagnostics-13-03640-f001:**
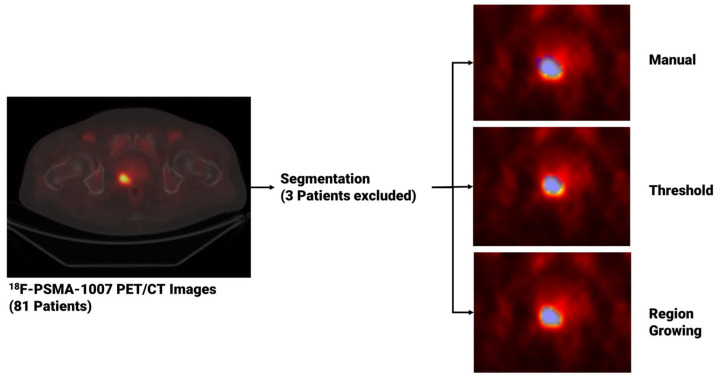
From image visualization to prostate cancer segmentation using manual, thresholding and region growing methods. In the image on the left, a PET slice displaying the tumor is visible, while on the right, three blue masks enclosing the prostate tumor are shown. These masks, obtained using three different segmentation algorithms, are then used to extract the radiomics features.

**Figure 2 diagnostics-13-03640-f002:**
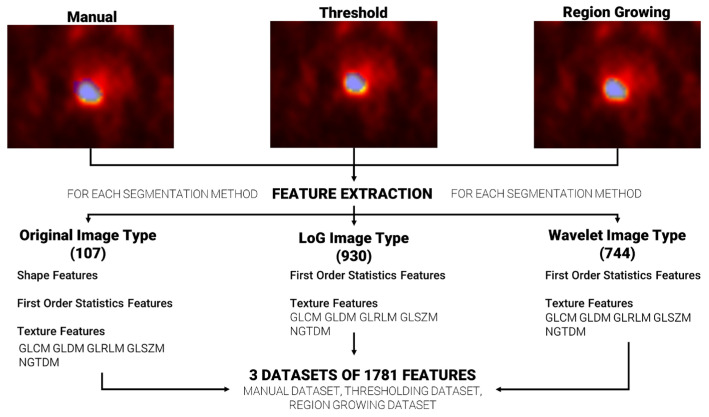
From segmentation to feature extraction for each segmentation method. The blue area that overlays each image represents the generated segmentation mask.

**Figure 3 diagnostics-13-03640-f003:**
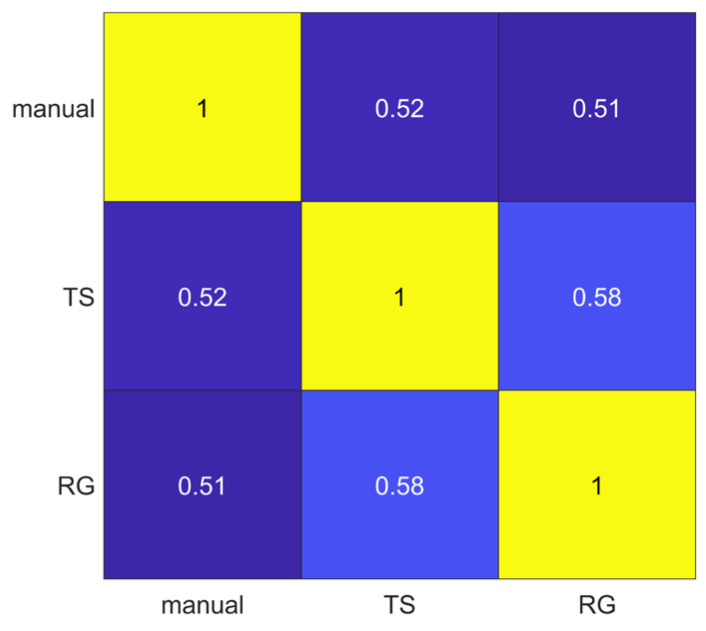
Pairwise segmentation agreement matrix between the three segmentation methods (manual, thresholding (TS) and region growing (RG). Numbers represent the average Jaccard Index values.

**Figure 4 diagnostics-13-03640-f004:**
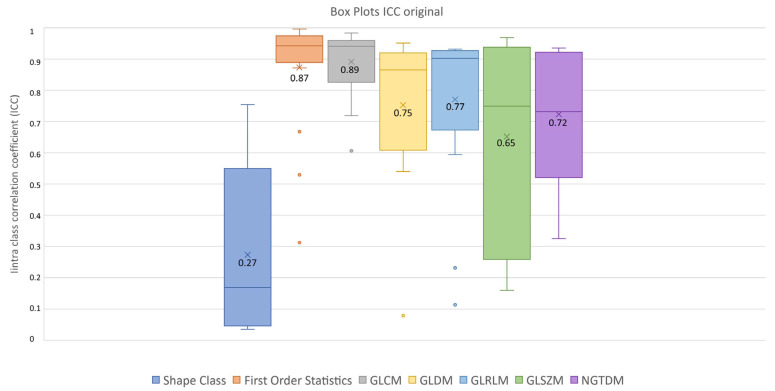
ICC box plots for the original image type. Mean value expressed by the black numbers next to the ‘×’ sign, while the horizontal line is the median.

**Figure 5 diagnostics-13-03640-f005:**
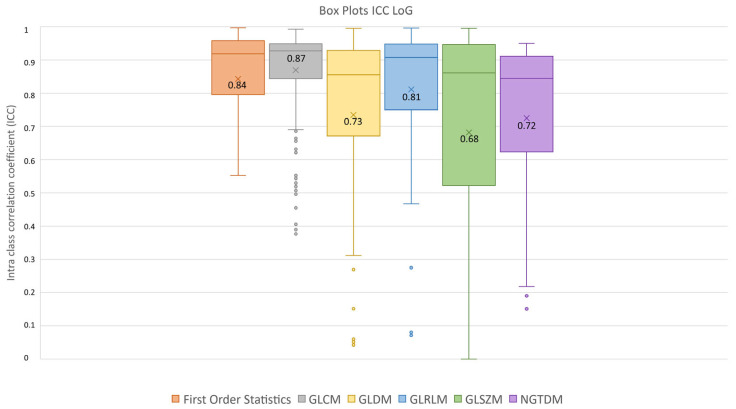
ICC box plots for the LoG image type. Mean value expressed by the black numbers next to the ‘×’ sign, while the horizontal line is the median.

**Figure 6 diagnostics-13-03640-f006:**
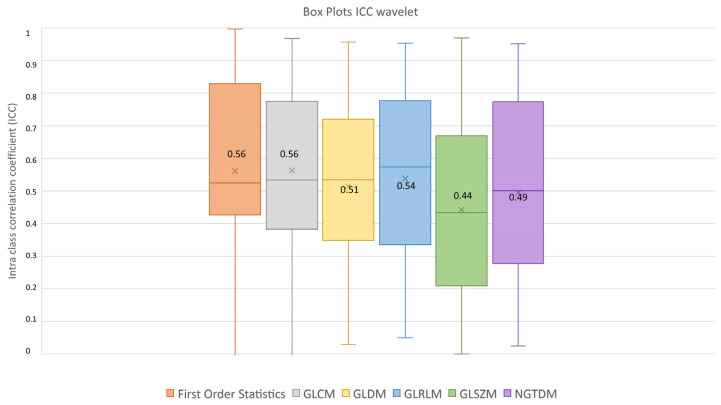
ICC box plots for the wavelet image type. Mean value expressed by the black numbers next to the ‘×’ sign, while the horizontal line is the median.

**Figure 8 diagnostics-13-03640-f008:**
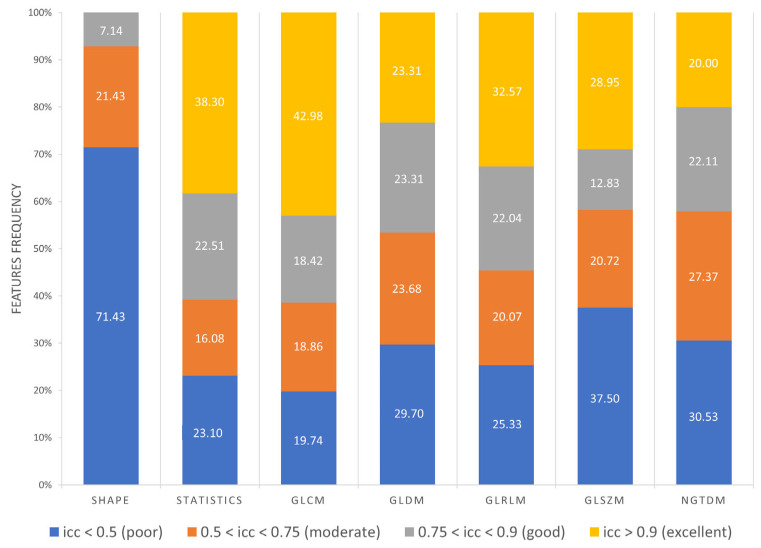
Frequency of poor, moderate, good, excellent reproducible features divided by feature class.

**Figure 9 diagnostics-13-03640-f009:**
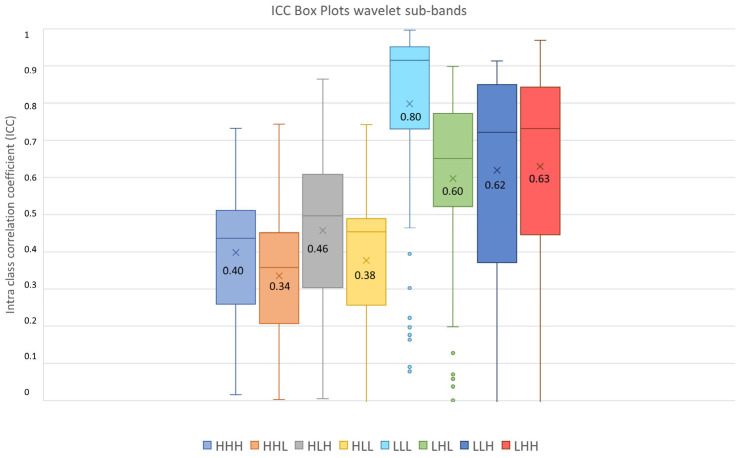
ICC box plots for the wavelet sub-bands (HHH, HHL, HLH, HLL, LLL, LHL, LLH, LHH). Mean value expressed by the black numbers next to the ‘×’ sign, while the horizontal line is the median.

**Table 1 diagnostics-13-03640-t001:** Pyradiomics configuration used for the feature extraction process.

Bin width	0.25
Isotropic Resampling	2 × 2 × 2
Interpolator	SitkBSpline
Wavelet Method	Coif1
Log Sigma	[0.5, 1, 1.5, 2, 2.5, 3, 3.5, 4, 4.5, 5]
Normalization	True; Scale = 1

**Table 2 diagnostics-13-03640-t002:** Characteristics of the 81 patients with primary prostate cancer involved in this study.

PET/CT Scanner	Patients	Low Grade GS = 3 + 3/3 + 4	High Grade GS = 4 + 3/4 + 4/4 + 5
GE	30	20	10
Siemens	51	26	25
Total	81	46	35

**Table 3 diagnostics-13-03640-t003:** Characteristics of the 78 patients after segmentation.

PET/CT Scanner	Patients	Low Grade GS = 3 + 3/3 + 4	High Grade GS = 4 + 3/4 + 4/4 + 5
GE	29	19	10
Siemens	49	24	25
Total	78	43	35

**Table 4 diagnostics-13-03640-t004:** Feature robustness results (average ICC) grouped by image type and feature class. The hyphen is present where shape features were not computed.

Image Type	Shape	Statistics	GLCM	GLDM	GLRLM	GLSZM	NGTDM
Original	0.27	0.87	0.89	0.75	0.77	0.65	0.72
LoG	-	0.84	0.87	0.73	0.81	0.68	0.72
Wavelet	-	0.56	0.56	0.51	0.54	0.44	0.49

**Table 5 diagnostics-13-03640-t005:** Subset of selected features divided by manual method.

Segmentation	Selected Features
Manual	*wavelet_LLL_firstorder_Minimum*
TS	*wavelet_HHL_glszm_LowGrayLevelZoneEmphasis*
RG	*wavelet_HLH_glszm_LowGrayLevelZoneEmphasis*

**Table 6 diagnostics-13-03640-t006:** LDA classifier performance for the three segmentation methods. Numbers in brackets represent the 95% confidence interval. Maximum values represented in bold.

Segmentation	Accuracy	AUC	Sensitivity	Specificity	Precision	Fscore
Manual	70.64% (70.47–70.81%)	73.08% (72.69–73.41%)	51.71% (51.43–52.19%)	**86.05%** (86.05–86.05%)	**75.10%** (74.99–75.27%)	61.25% (61.01–61.63%)
TS	70.60% (70.26–70.85%)	73.40% (73.63–74.38%)	74.48% (74.29–74.95%)	67.44% (66.90–67.91%)	65.07% (64.69–65.42%)	69.45% (69.21–69.72%)
RG	**74.49%** (74.10–74.91%)	**79.20%** (78.89–79.50%)	**84.38%** (83.90–84.86%)	66.43% (65.81–67.13%)	67.19% (66.78–67.65%)	**74.80%** (74.44–75.19%)

**Table 7 diagnostics-13-03640-t007:** SVM classifier performance for the three segmentation methods. Numbers in brackets represent the 95% confidence interval. Maximum values represented in bold.

Segmentation	Accuracy	AUC	Sensitivity	Specificity	Precision	Fscore
Manual	70.47% (70–71.11%)	73% (72.56–73.41%)	57.05% (56.29–58%)	**81.40%** (80.78–82.17%)	**71.43%** (70.73–72.45%)	63.41% (62.72–64.17%)
TS	68.50% (67.78–69.19%)	72.50% (71.89–73.08%)	74.19% (73.43–74.86%)	63.88% (63.02–64.81%)	62.60% (61.91–63.31%)	67.89% (67.19–68.51%)
RG	**71.84%** (74.10–74.91%)	**77.32%** (77.06–77.63%)	**85.24%** (84.57–85.71%)	60.93% (60.16–61.63%)	63.99% (63.57–64.38%)	**73.09%** (72.71–73.40%)

**Table 8 diagnostics-13-03640-t008:** KNN classifier performance for the three segmentation methods. Numbers in brackets represent the 95% confidence interval. Maximum values represented in bold.

Segmentation	Accuracy	AUC	Sensitivity	Specificity	Precision	Fscore
Manual	70.26% (69.65–71.07%)	67.41% (66.76–67.97%)	58.76% (57.62–59.71%)	**79.61%** (78.76–80.70%)	**70.19%** (69.25–71.39%)	63.92% (63.04–64.77%)
TS	66.37% (65.62–67.14%)	68.58% (67.84–69.18%)	67.33% (66.19–68.38%)	65.58% (64.49–67.21%)	61.51% (60.61–62.43%)	64.23% (63.51–65.03%)
RG	**71.75%** (70.81–72.61%)	**77.86%** (77.13–78.58%)	**77.05%** (76–78.10%)	67.44% (66.12–68.84%)	65.91% (64.89–66.87%)	**71%** (70.13–71.82%)

**Table 9 diagnostics-13-03640-t009:** RF classifier performance for the three segmentation methods. Numbers in brackets represent the 95% confidence interval. Maximum values represented in bold.

Segmentation	Accuracy	AUC	Sensitivity	Specificity	Precision	Fscore
Manual	67.65% (66.88–68.46%)	66.77% (65.79–67.61%)	57.81% (56–59.33%)	**75.66%** (74.73–76.59%)	**65.90%** (64.93–66.98%)	61.52% (60.18–62.66%)
TS	63.55% (62.52–64.66%)	67.40% (66.55–68.12%)	65.62% (63.60–67.81%)	61.86% (60.19–63.68%)	58.40% (57.43–59.62%)	61.67% (60.29–63.03%)
RG	**71.15%** (70.43–72.01%)	**73.82%** (73.01–74.61%)	**77.05%** (75.62–78.48%)	66.36% (65.27–67.37%)	65.12% (64.44–65.94%)	**70.54%** (69.72–71.46%)

**Table 10 diagnostics-13-03640-t010:** AdaBoost classifier performance for the three segmentation methods. Numbers in brackets represent the 95% confidence interval. Maximum values represented in bold.

Segmentation	Accuracy	AUC	Sensitivity	Specificity	Precision	Fscore
Manual	63.16% (61.79–64.44%)	62.91% (61.81–63.88%)	54.48% (52.27–56.57%)	**70.23%** (68.45–71.71%)	**59.87%** (58.19–61.35%)	56.95% (55.09–58.56%)
TS	57.22% (56.11–58.33%)	62.70% (61.56–63.74%)	58.19% (56.10–60.38%)	56.43% (54.88–57.91%)	52.06% (50.96–53.32%)	54.87% 53.51–56.30%)
RG	**63.72%** (62.56–64.74%)	**68.99%** (67.88–70.05%)	**64.19%** (61.71–65.71%)	63.33% (61.86–64.81%)	58.78% (57.52–59.81%)	**61.29%** (59.64–62.40%)

**Table 11 diagnostics-13-03640-t011:** NN classifier performance for the three segmentation methods. Numbers in brackets represent the 95% confidence interval. Maximum values represented in bold.

Segmentation	Accuracy	AUC	Sensitivity	Specificity	Precision	Fscore
Manual	58.72% (57.48–60.56%)	56.05% (54.20–57.97%)	46.19% (43.43–48.95%)	**68.91%** (66.82–70.68%)	**54.69%** (52.90–57.02%)	49.90% (47.79–52.48%)
TS	62.14% (60.66–63.50%)	63.50% (61.91–65.04%)	66.76% (64.57–69.14%)	58.37% (56.43–60.54%)	56.74% (55.34–58.23%)	61.22% (59.70–62.74%)
RG	**65.77%** (64.27–66.97%)	**69.24%** (67.96–70.33%)	**69.14%** (65.90–71.52%)	63.02% (61.32–64.73%)	60.35% (58.95–61.55%)	**64.29%** (62.18–65.79%)

## Data Availability

Data are available for bona fide researchers who request it from the authors.

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
