# Peer review of "A Critical Analysis of the Robustness of Radiomics to Variations in Segmentation Methods in 18F-PSMA-1007 PET Images of Patients Affected by Prostate Cancer"

_diagnostics, 2023, doi:10.3390/diagnostics13243640_

Round 1
Reviewer 1 Report
Comments and Suggestions for Authors
Summary
The authors study robustness of radiomics to variations in segmentation methods in 18F-PSMA-1007 PET images of pa- tients affected by prostate cancer. The topic is important and interesting. However, the analysis needs to be deepened and below in major comments are suggestions to do this. After carefully considering and addressing the comments below, the manuscript could be published in Diagnostics.
Major comments
-It would be interesting and relevant to consider the results of segmentation before feature extraction. How does the binary segments from different methods compare using e.g. Dice score or Jaccard index? Please add this type of analysis.
-It is unclear why only 3 traditional ML methods were tested. Please justify or include at least random forest, boosting and neural networks.
-Please add statistical analysis (=suitable statistical test to obtain P-values) when comparing numeric results such as ICC, accuracy, AUC, etc. Perhaps after this analysis Discussion also needs to be revised.
-For results in section 3.2 please show the results per iteration and make statistics over the iterations to see the variation. Compare sensitivity & specificity (e.g. McNemar’s test) as a pair, and preferably add also F1 score as a metric.
-Explainability is a problem in the radiomics approach. Please discuss this issue.
Minor comments
-Page 2, last line, and page 3 first line: Please uniformise your notations. Now you have “128*128” and “512x512”.
-Page 3, line 98: It is unclear is the given voxel size for PET or CT. Please clarify.
-Page 3, line 102: Please clarify is the given voxel size for PET, CT or both.
-Page, 3, line 102: In “mm3” number 3 should be superscript.
-Page 3, lines 109-112: It would be good for the reader to mention already here how many images per scanner there are, and how many were excluded. (Or refer to the table, which contains this information.)
-Page 128-129: It is unclear what happens to the other scores and why you selected these to low and high. Would it be better to have a clear threshold? In that case, all possible gradings can be classified into low and high. (While you only have these, someone else could also apply same methodology and have also different values.)
-Page 3, line 134: Please elaborate on “manual”. What does it mean? Pixel by pixel?
-Page 3, line 134: Please elaborate on “region-growing based”. What does it mean? Which method?
-Page 3, line 140: Why could not be segmented? Please clarify (and show example figure, if possible).
-Page 5, Fig. 2: Font size of some texts is too small.
-Page 5, line 180: Please define ICC variables (ICC_shape, ICC_Statistics,…) so that reader does not have to guess their meaning.
-Page 5, lines 184-186: These explanations should be right after the related formula.
-Page 6, line 202: Better to say “5-fold” instead of “k-fold”, if that is what you did.
-Page 6, line 202: Please clarify, did you use the same folds for each method in all of the 10 iterations? (You should have, because otherwise results are not comparable.)
-Page 6, lines 220-228: It would be beneficial to compare the results using suitable statistical test.
-Box plots from pages 7-8 and 10: Please clarify that the line is median and the “x” with numeric value is mean (if I have understood it correctly).
-Box plots from pages 7-8: Some of the text is in too small fonts, especially the legend. Are all Figs. 3-5 needed? Could they be combined or some of them moved to supplement?
-Box plots from pages 9-10: Some of the text is in too small fonts, especially the legend.
Author Response
Dear Reviewer,
we attached a .pdf file with the answers to your comments.
Best regards,
the Authors

Reviewer 2 Report
Comments and Suggestions for Authors
This work demonstrate analysis of radiomics in prostate cancer segmentation in PET imaging. The manuscript is well-written and the contribution has some originality related to the tracer. Minor changes are need before manuscript is ready for publication.
1) In medical imaging, image segmentation usually means labeling images to identify specific region or abnormality. However, in illustrations shown here (for example, Fig. 1), It is unclear how segmentation is visualized. Please clarify or edit figure(s) as needed.
2) Validity of presented results need to confirmed through the ability of reproduction. As the data were not shared, it is difficult to validate the results presented here. Moreover, there is no comparison with any potential earlier work that is needed to understand how the results here standing.
3) In discussion, authors stated that "From our study emerged that segmentation methods influence radiomics features and Shape features are negatively affected", which is obvious in my opinion.
4) Emphesis originality is important, but authors repeat the "To the best of our knowledge..." statement several times in Introduction, discussion and conclusions. There are several important studies already out there that discuss the same problem and the scope of data here is very limited. Please adjust the contribution here to what it should be.
Author Response

(The authors gave the same response as above.)

Reviewer 3 Report
Comments and Suggestions for Authors
Dear authors,
My review is attached.
Best regards,

Author Response

(The authors gave the same response as above.)

Round 2
Reviewer 2 Report
Comments and Suggestions for Authors
I have no further comments.